# Application of Monoclonal Antibodies against Naturally Occurring Bioactive Ingredients

**DOI:** 10.3390/antib13030060

**Published:** 2024-07-24

**Authors:** Shunsuke Fujii, Takuhiro Uto, Hiroaki Hayashi, Waraporn Putalun, Seiichi Sakamoto, Hiroyuki Tanaka, Yukihiro Shoyama

**Affiliations:** 1Faculty of Pharmaceutical Sciences, Nagasaki International University, 2825-7 Huis Ten Bosch, Sasebo 859-3298, Nagasaki, Japan; fujii@niu.ac.jp (S.F.); uto@niu.ac.jp (T.U.); 2College of Pharmaceutical Sciences, Ritsumeikan University, 1-1-1 Nojihigashi, Kusatsu 525-8577, Shiga, Japan; hhayashi@fc.ritsumei.ac.jp; 3Faculty of Pharmaceutical Sciences, Khon Kaen University, Khon Kaen 40002, Thailand; waraporn6@yahoo.com; 4Graduate School of Pharmaceutical Sciences, Kyushu University, 3-1-1 Maidashi, Higashi-ku, Fukuoka 812-8582, Fukuoka, Japan; s.sakamoto@phar.kyushu-u.ac.jp; 5Faculty of Pharmaceutical Sciences, Sanyo-Onoda City University, 1-1-1 Daigaku-dori, Yamaguchi 756-0884, Yamaguchi, Japan; htanaka@rs.socu.ac.jp

**Keywords:** monoclonal antibody, eastern blotting, knockout extractives, pharmacological activity, Kampo medicine

## Abstract

Monoclonal antibodies (Mabs) are widely used in a variety of fields, including protein identification, life sciences, medicine, and natural product chemistry. This review focuses on Mabs against naturally occurring active compounds. The preparation of Mabs against various active compounds began in the 1980s, and now there are fewer than 50 types. Eastern blotting, which was developed as an antibody staining method for low-molecular-weight compounds, is useful for its ability to visually represent specific components. In this method, a mixture of lower-molecular-weight compounds, particularly glycosides, are separated by thin-layer chromatography (TLC). The compounds are then transferred to a membrane by heating, followed by treatment with potassium periodate (KIO_4_) to open the sugar moiety of the glycoside on the membrane to form an aldehyde group. Proteins are then added to form Schiff base bonds to enable adsorption on the membrane. A Mab is bound to the glycoside moiety on the membrane and reacts with a secondary antibody to produce color. Double Eastern blotting, which enables the simultaneous coloration of two glycosides, can be used to evaluate quality and estimate pharmacological effects. An example of staining by Eastern blotting and a component search based on the results will also be presented. A Mab-associated affinity column is a method for isolating antigen molecules in a single step. However, the usefulness of the wash fractions that are not bound to the affinity column is unknown. Therefore, we designated the wash fraction the “knockout extract”. Comparing the nitric oxide (NO) production of a glycyrrhizin (GL)-knockout extract of licorice with a licorice extract revealed that the licorice extract is stronger. Therefore, the addition of GL to the GL-knockout extract of licorice increased NO production. This indicates that GL has synergic activity with the knockout extract. The GL-knockout extract of licorice inhibited high-glucose-induced epithelial–mesenchymal transition in NRK-52E cells, primarily by suppressing the Notch2 pathway. The real active constituent in licorice may be constituents other than GL, which is the causative agent of pseudohyperaldosteronism. This suggests that a GL-knockout extract of licorice may be useful for the treatment of diabetic nephritis.

## 1. Introduction

Natural product research has become more extensive, more sophisticated, and more developed. One reason is that the goal is to develop drugs from natural products. However, because the majority of medicinal plants that are used for natural medicines are derived from nature, thus, heterogeneity in their quality is a major problem. As a result, the development of methods to evaluate quality and quality control has been conducted for a long time. Furthermore, the diversification of medical care in Japan and the distribution of Kampo medicines have increased recently. More than 80% of the prescribed herbal medicines are imported and 70% are from wild species. Therefore, it is impossible to keep the medicinal effects of crude drug preparations constant unless the homogeneity of the crude drug ingredients is guaranteed. For this reason, the quality of crude drugs is strictly regulated by the Japanese Pharmacopoeia. The number of medical Kampo prescriptions listed in the 18^th^ edition of the Japanese Pharmacopoeia revised in 2021 is 37 out of 148 prescriptions. In 2021, the number of Kampo prescriptions listed in medical guidelines was 112 prescriptions. This indicates that the usefulness of Kampo prescriptions in modern medicine has been recognized and we have entered a very important period of change. Thus, research on the standardization of herbal medicines has become more important. Most of the conventional analysis methodologies are based on high-performance liquid chromatography (HPLC) [1,2,3]. This requires expensive machinery and separation columns, which must be replaced before they deteriorate. Moreover, the columns are usually replaced with an organic solvent as the eluent. The analytical methods are expensive and require a lot of work. In contrast to conventional analytical methods, the analysis of medicinal ingredients using monoclonal antibodies (Mabs) does not require pretreatment, such as preliminary purification. The sensitivity of this method is 100- to 1000-fold higher compared with that of conventional methods and the reproducibility is excellent. In addition, this method does not require organic solvents, making it an eco-friendly analytical method. Under these circumstances, the unique analytical methods using Mabs with the most important natural gradients, ginseng, and licorice will be discussed in this review. The *Panax* genus, which belongs to the family Araliaceae, consists of eleven species, nine of which grow in Asia and two in North America [4,5]. Of these, *P. ginseng*, *P. notoginseng*, and *P. quinquefolius* are cultivated in Asia and North America and are three major species in the worldwide medicinal product market. These three species primarily contain ginsenosides and 257 ginsenosides have been isolated from ginseng alone [6,7]. Figure 1 shows the biosynthetic pathway of ginsenosides [8]. Protopanaxadiol and protopanaxatriol are two characteristic triterpenoid saponins in ginsenosides. All ginsenosides belong to either the protopanaxadiol or protopanaxatriol series. As shown in the biosynthetic pathway, the quality control of ginseng may be achieved by establishing an immunochemical evaluation system targeting ginsenosides, such as protopanaxadiol and protopanaxatriol.

Licorice contains 500 or more compounds. Of these, glycyrrhizin (GL) has a higher concentration and has been used to treat hepatitis and allergies in Japan for nearly 50 years. Licorice is also an important herb used in 70% of Kampo medicines in Japan. However, there are well-known side effects of licorice, such as hypokalemia or pseudohypoaldosteronism [9]. For this reason, the Japanese Pharmacopoeia specifies a GL content of around 2% for licorice. Nonetheless, licorice contains many flavonoids and isoflavone compounds. The biosynthesis of GL and flavonoids is outlined below. The biosynthesis of GL has been well studied and most of the genes involved have been identified. A common triterpene intermediate β-amyrin is oxidated at C-11 by CYP88D6 [10] and at C-30 by CYP72A154 [10] to produce glycyrrhetinic acid (GA), the aglycone of GL. GA is further glycosylated by UDP-glycosyltransferases (UGTs). The glucuronylation of GA is catalyzed by GuUGAT [11], UGT73F15 [12], and GuCSyGT [13]. GuCSyGT is responsible for soyasaponin biosynthesis by disrupting the GuCSyGT gene in licorice hairy roots [14]; however, it is necessary to confirm the gene responsible for the glucuronidation of GA during GL biosynthesis in *Glycyrrhiza* plants. The biosynthesis pathway of liquiritin (LQ) has also been well characterized. Three molecules of malonyl-CoA and 4-coumaroyl-CoA, synthesized from L-phenylalanine, are condensed by chalcone synthase (CHS) and polyketide reductase (chalcone reductase) to produce 6′-deoxychalcone isoliquiritigenin [15]. Isoliquiritigenin is isomerized into liquiritigenin, 5-deoxyflavanone, by chalcone isomerase (CHI) [16]. Next, liquiritigenin is glycosylated to form LQ, a flavanone glycoside, by UGTs. The glucosylation of liquiritigenin to LQ is catalyzed by several UGTs in licorice [17]. Both GL and LQ selectively accumulate in the woody parts of the tap roots and stolon of licorice [18,19]. The GL content is positively correlated with the LQ content in the tap roots among 100 *Glycyrrhiza* plants [20]. In addition, the GL and LQ contents increase simultaneously by salt stress [21] and arbuscular mycorrhizal symbiosis [22]. The biosynthetic enzymes for GL (CYP88D6 and CYP72A154) and LQ (CHS and CHI) are also upregulated by salt stress and arbuscular mycorrhizal symbiosis [22], suggesting that the biosynthesis of GL and LQ share a common regulatory mechanism. Further experiments are needed to elucidate the regulation of GL and LQ biosynthesis. Metabolites of GL have been implicated in the side effects of GL and metabolic studies and mechanisms of action have been conducted. GL inhibits type 2 11β-hydroxysteroid dehydrogenase, which degrades cortisol to inactive cortisone and induces mineral corticoid receptor activity. Under these conditions, metabolite studies of GL have been performed, and GA, its monoglucuronide, and its sulfate are well known [9].

## 2. Monoclonal Antibody (Mab)

The history of antibodies against natural products indicates that in the 1970s, polyclonal antibodies (Pabs) were the standard reagent; however, in the 1980s, Mabs began to be produced, but they were limited to the measurement of morphine, which is medicine. One of the reasons was the lack of a method to accurately measure the number of natural products bound to the carrier proteins in the antigen. The introduction of mass spectrometry and matrix-assisted laser desorption/ionization–tof–mass spectrometry (MALDI-TOF-MS) enabled the accurate determination of the number of bound natural products in the hapten and the reliable preparation of Mabs [23]. As the techniques for Mab creation improved and the need for Mabs increased, Mabs have been generated for various pharmacologically active natural products and there are now more than 40 species.

Most Mabs are selected for their low cross-reactivity and specific binding to antigenic molecules, although Mabs with broad cross-reactivity may be selected for specific purposes. For example, Mabs have been used to ensure quality control of crude drugs by a quantitative analysis of various natural products [24]. The sensitivity of Mabs is usually 10- to 100-fold higher compared with HPLC. Normally, an enzyme-linked immunosorbent assay (ELISA) is used for these analyses, but surface plasmon resonance-based immunosensors are used for more efficient analyses. Furthermore, formononetin, which is distributed widely in the Leguminosae family, has been isolated from a licorice extract as an in vitro fertilization activator for mouse sperm; thus, an anti-formononetin Mab was generated [25]. A method was developed to analyze formononetin using an indirect competitive ELISA with the anti-formononetin Mab. This assay may be used in a cell-based assay system using MC3T3-E1 cells because the concentration of formononetin and its glycoside, ononin, taken up by MC3T3-E1 cells may be quantified [25].

Tetrahydrocannabinolic acid (THCA) biosynthetic enzyme studies have been conducted [26] as well as metabolic studies [27]. Although Eastern blotting described in detail below has not been performed, an anti-THCA Mab, which has cross-reactivity to all cannabinoids, was developed [27]. Metabolic studies of tetrahydrocannabinol (THC) are also important in drug discovery, as its bioavailability differs depending on its structure and affinity to enzymes and/or receptors [28]. Mabs can discriminate between various THC metabolites with different oxidation states, but they can also distinguish between stereoisomers as well. Because this Mab does not react with endocannabinoids, cholesterol, testosterone, or β-carotene, it may be used for kinetic studies of THC [27]. In vitro and in vivo studies indicate that rapid THC oxidation occurs at various sites on the THC molecule, and THC metabolites, in which hydroxyl, epoxy, carbonyl, and carboxylic acid groups are introduced by oxidation, have been isolated and structurally elucidated. The metabolic pathway is shown in Figure 2. Metabolic studies of THC are also important for drug discovery, as its bioavailability differs depending on its structure, and its affinity to enzymes and/or receptors is also greatly affected. As mentioned, Mabs can discriminate between various THC metabolites with different oxidation states; however, they can also discriminate between stereoisomers. The β-isomer of 7-hydroxy-Δ^8^-THC and 8-hydroxy-Δ^9^-THC was found to be slightly more reactive than the α-isomer; however, the double-bond isomers could not be distinguished between the two. Recently, Elmes et al., reported that fat-soluble THC is transferred by fatty acid-binding proteins [29]. Subsequently, a pathway was identified, in which THC is transferred and stored in the ligand-binding pocket by fatty acid-binding proteins before being transferred to CYP450 enzymes for metabolism [30]. This suggests that newly prepared anti-THCA Mabs will be useful for the above studies to track the movement of THC metabolites in vivo. As we describe in detail below, a Mab against solamargine, a steroidal alkaloid saponin, also exhibits broad cross-reactivity against most solasodine saponins [31].

Eastern blotting [32] is a method of coloration for small molecules, similar to Western blotting. It is essential for peptide and/or protein research and has expanded the scope of research on natural products. Furthermore, immunoaffinity columns prepared with Mabs for low-molecular-weight natural products were developed and a one-step method was established to isolate antigen molecules. It became possible to prepare an extract, in which only the antigenic molecules were removed from the crude extract, and this extract was designated a “knockout extract” [33].

Meanwhile, single-chain variable regions of heavy (V_H_) and light (V_L_) (scFv) antibody genes for the variable portion of Mabs were also created because the management of hybridomas that produce Mabs requires significant time and cost. Heavy V_H_ and light V_L_ antibody genes were directly cloned from the cDNA of hybridomas and assembled using splicing by overlap extension PCR with linker DNA. The constructed scFv genes were cloned and the recombinant scFv exhibited cross-reactivity as its mother Mab. Therefore, the newly expressed scFv protein can be used for quantitative ELISA for the determination of antigen molecule concentration in natural products. The development of scFv protein chemistry spurred the development of antibody drugs, with more than 100 now on the market. ScFv antibodies have also been used widely for biotechnological approaches [34] since they have the same monovalent binding specificity and affinity as their parent antibodies [35]. In the field of plant pathology, scFvs have been used as a diagnostic tool [36,37,38]. Moreover, scFv can be applied for the investigation of plant growth [39]. Furthermore, regarding plant biotechnology, a new breeding method known as “missile-type molecular breeding” was developed, which was aimed at breeding species with high antigenic molecule content. The anti-solamargine Mab, which has wide affinity for solasodine glycosides, is also a specific Mab that has contributed to the breeding of species containing high levels of solasodine glycosides by introducing the scFv antibody gene [31]. This is the first success of molecular breeding for secondary metabolites without treatment of a biosynthetic enzyme.

As described above, Mab creation and its application for natural products have expanded as discussed in detail below.

## 3. Eastern Blotting

Among the immunochemical staining methods using Mabs, Western blotting is an essential method for proteins and peptides in biological research. However, because there is no method for lower-molecular-weight compounds, the authors developed a method to detect such compounds on membranes using a Mab in a three-step process known as Eastern blotting [32]. The following is a brief overview. Low-molecular-weight compounds are developed and separated by TLC (Step 1) and compounds on TLC are transferred to polyvinylidene fluoride (PVDF) or polyethersulphone (PES) by heating (Step 2). The blotted compounds on the membrane are easily washed away. The remaining compounds on the membrane are cleaved by NaIO_4_ to yield an aldehyde-type intermediate with a carrier protein fixed on the membrane (Step 3) and then stained (Step 4). In this state, the glycosides are chromogenized by conventional Western blotting using Mabs. The disadvantage of this method is that the affinity of the Mab for the sugar moiety is reduced and only recognizes the aglycone moiety because the glycosides are modified with NaIO_4_ to form an aldehyde group to create a conjugate through Schiff bonding with the protein. The unique saponins of ginseng, the ginsenosides, are classified into protopanaxadiol and protopanaxatriol depending on the type of aglycone [40]. In Eastern blotting, protopanaxadiol and protopanaxatriol may be grouped and dyed according to the Mabs for each. This indicates that the affinity of the sugar moiety has been lost, although it is considered a useful method for the quality control of ginseng. This methodology has been applied to naturally occurring compounds, such as sennosides in *Rheum*, ginsenosides in *Panax*, and GL in *Glycyrrhiza* spp. For components other than glycosides, it is necessary to devise a way to fix them to the membrane. For example, in the case of aristolochic acid, the carboxylic acid portion is reacted with N-hydroxysuccinimide to bind to the protein via an intermediate in the membrane [41] (Figure 3).

### 3.1. Single Eastern Blotting

Three types of natural products, solasodine glycosides, ginsenosides, and GL, are discussed in this section. Solasodine glycosides are an example of staining for almost all solasodine glycosides using a Mab with wide cross-reactivity. For example, the anti-solamargine Mab recognizes all solasodine glycosides [42]. In this case, all solasodine glycosides are hydrolyzed to yield solasodine, which is used as a reagent in biochemical studies. For example, solasodine reduces PI3K/Akt signaling pathways and downregulates miR-21 expression [43].

Because ginsenoside Rb1 is a major component of ginseng and has important pharmacological activities [44,45], various species belonging to the Araliaceous family have been analyzed by ELISA and single Eastern blotting using the anti-ginsenoside Rb1 Mab (see Table 1). *Kalopanax pictus* and *Fatsia japonica* were positive in ELISA and Eastern blotting; however, *F. japonica* showed weak coloration in the Eastern blot experiments. From the double surveys, it became evident that *K. pictus* may contain ginsenoside Rb1. Compared with the concentration of barks and leaves by ELISA, the bark extract was purified using various types of column chromatography to yield ginsenoside Rb1 [46].

In *P. japonicus* grown in Japan, oleanane saponin is a major component and ginsenosides are present in small amounts [47]. Therefore, it is extremely difficult to detect ginsenosides mixed in trace amounts among a large amount of oleanane saponin. Tanaka et al., performed Eastern blotting on a *P. japonicus* extract and successfully detected two spots, although the content of compound **1** appeared to be minuscule. To confirm that compounds **1** and **2** were protopanaxadiol-type ginsenosides, an anti-ginsenoside Rb1 Mab-conjugated affinity column was prepared. The *P. japonicus* extract was charged to the affinity column, washed thoroughly, and eluted. The results indicated that the two spots were separated and stained by the anti-ginsenoside Rb1 Mab. The coloration of these two saponins indicated that the aglycones were protopanaxadiol, and the number of bound sugars was predicted to be three and five, respectively, based on the Rf values. Comparing the Rf values of ginsenoside Rd, Rc, and Rb1 with that of compound **1** revealed that compound **1** was closer to ginsenoside Rd with three sugars per molecule. Therefore, the number of sugars in compound **1** was determined to be three. However, the Rf of compound **2** was lower compared with that of ginsenoside Rb1 containing four sugars, which suggests that the sugar number in compound **2** was five. Based on the literature [48], compound **1** was presumed to be chikusetsusaponin III, which could be identified by direct comparison with the sample. Compound **2** was confirmed to be chikusetsusaonin VI by direct comparison. The elution pattern shows that compound **1** is eluted by the wash solution. This indicates that the affinity of the column is not very strong. Simultaneously, compound **2** was eluted with an eluant, indicating that it was fixed onto the column, but readily eluted with the eluent. This shows that ginsenosides can be separated and purified based on the number of bound sugars by controlling the affinity of the column. Moreover, because the type of bound sugar influences the Rf value, the combination of Eastern blotting and the affinity column elution pattern is useful to confirm the structure of unknown ginsenosides. This is a new finding for natural product chemistry, and the authors expect that it will be applied to a wider range of natural products in the future.

Cases of pseudoaldosteronism have been reported with long-term use of high-dose GL [9,49,50]. Therefore, a simple and quick method to measure GL is desired; thus, Eastern blotting was developed. GL was injected intravenously into mice and serum was analyzed over time as shown in Figure 4. It is impossible to detect GL in the serum by TLC. However, GL is readily detectable by Eastern blotting. Because GL is present in the blood for approximately two hours, the metabolic rate was assumed to be fast. From this experiment, it is clear that GL concentrations can be measured in the serum by Eastern blotting (Figure 4).

Another example shows the effectiveness of Eastern blotting [32]. Approximately 70% of Kampo medicines are prescribed licorice [51], which indicates that licorice is an important herbal medicine. Moreover, to reduce the side effects of GL, the daily limit of licorice was established at 6 g in Japan. Therefore, it is necessary to know the approximate amount of GL in the Kampo prescription. Using TLC, it is impossible to detect GL from the chromatogram of the Kampo prescription; however, with Eastern blotting, it is possible to detect GL in Kampo medicine with licorice. In the Kampo prescription without licorice, no GL spots were observed [32]. These results indicate that Eastern blotting can be applied as an easy and convenient method to test whether a Kampo prescription contains licorice.

There is an example of the identification of the GL metabolite, GA sulfate, using an anti-GA monoglucronide Mab by Eastern blotting, whereas research on the metabolites is ongoing [52].

### 3.2. Double Eastern Blotting

The authors developed a new assay system, double Eastern blotting, for ginsenosides in ginseng, GL and LQ in licorice, and sennoside A and B in rhubarb [53]. The double Eastern blotting method can be divided into two functional fields. The first works when components of the same group have different steric structures or different functional groups in a molecule. The second is an example of the simultaneous detection of completely different components. The first example is a staining method for ginsenosides contained in ginseng that can select for the presence or absence of a hydroxyl group at a sterically important position. This difference in molecular structure is also known to have varying pharmacological activity, such as anti-cancer activity [54]. The structural difference (i.e., protopanaxadiol-type ginsenoside Rg3) has been widely studied for its strong antitumor activity [48] and developed as an anti-cancer drug in China. Protopanaxatriol-type ginsenosides can enhance the excitability of the hippocampus, resulting in enhanced memory and learning ability [55]. Based on these results, it is possible to predict the pharmacological activity of Panax spp. By determining whether the ginsenosides contained in the genus Panax are mostly protopanaxadiol or protopanaxatriol. This information is also important for quality control [56]. Therefore, a methodology that can easily distinguish between both ginsenosides for quality control is desirable. The authors have succeeded in double Eastern blotting with two Mabs, the anti-ginsenoside Rb1 (protopanaxadiol-type) Mab and anti-ginsenoside Rg1 (protopanaxatriol-type) Mab (Figure 5).

A method was devised to dye two groups of components at different hydroxyl group positions for ginsenosides using two MAbs. Ginsenosides in ginseng are divided into two groups related to the 6-position on a dammarane skeleton, such as protopanaxadiol (a) and protopanaxatriol (b) groups (Figure 5). Two MAbs against ginsenoside Rb1 (no hydroxy group at C6) and Rg1 (a hydroxyl group at C6) were prepared because protopanaxadiol-type ginsenosides and protopanaxatriol-type ginsenosides have different pharmacological activities, respectively. It became evident that protopanaxadiol- and protopanaxatriol-type ginsenosides can be detected by blush and reddish coloring, respectively. The profiles of various ginseng components were elucidated by Eastern blotting and double Eastern blotting using anti-ginsenoside Rb1 and Rg1 MAbs. Comparing the sulfuric acid and Eastern blotting profiles revealed that the Rf values for each spot were the same, but the intensity patterns of the spots were different. In addition, while there is no variation in coloration by spot in the sulfuric acid profile, in the Eastern blot profile, some spots are blush, some are reddish, and some are a mixture of both colors. Depending on the individual spot, blush and reddish spots showed protopanaxadiol- and protopanaxatriol-type ginsenosides, respectively. Spots in both were mixed and were also recognized [53].

Because the affinity of the sugar moiety was lost, but the aglycone moiety was still detectable by Eastern blotting, two types of MAbs for ginsenoside, ginsenoside Rb1, and ginsenoside Rg1 with different aglycone moieties, protopanaxadiol and protopanaxatriol, were generated. When Eastern blotting was performed, all protopanaxadiol family ginsenosides, such as ginsenoside Rb1, Rb2, Rc, and Rd, were chromogenic. However, when the protopanaxatriol-based anti-ginsenoside Rg1MAb was used, ginsenoside Re, Rf, and Rh1 were colored. There are several advantages of double Eastern blotting. For example, coloring shows pharmacological activity for the quality control of ginseng and/or the detection of the *Panax* species. One question is why many protopanaxadiol-type ginsenosides stain blue and most protopanaxatriol-type ginsenosides stain reddish, even though the cross-reactivity of the anti-ginsenoside Rb1 MAb against corresponding ginsenosides was small. This suggests that an aglycone, protopanaxadiol, and its sugars may be useful for immunization and may function as an epitope for the structure of ginsenosides. In addition, it was suggested that the specific reactivity of the sugar moiety in the ginsenoside molecule against the anti-ginsenoside Rb1 MAb may be modified by cutting the sugar moiety by NaIO_4_ treatment of ginsenosides on the PVDF membrane. This enables the detection of ginsenosides Rc and Rd by Eastern blotting. In this method, cross-reactivity weakened and ginsenosides belonging to protopanaxadiol and protopanaxatriol were detected in two groups. Therefore, each band may be identified as belonging to either protopanaxadiol or protopanaxatriol by Eastern blotting, which results in a prediction of the pharmacological activity of each ginsenoside [57]. A second advantage is that in Eastern blotting, the Rf value of TLC is transferred to the membrane as previously indicated. Thus, the number of sugars can be inferred from the Rf value. Because the type of aglycone can be confirmed from the coloration of the previous band and the number of sugars can be determined from the Rf value, an estimated structural formula can be generated and the matching ginsenoside can be identified from the database [58]. There is one interesting aspect about glycosides, which is the more sugars in the same aglycone, the more active it is. Because the number of sugars can be estimated by Eastern blotting, the degree of activity can be readily estimated. This tendency was found in the biologically active natural saponins, including ginsenosides [59], saikosaponins [60], and cardiac steroids [61].

Next, we present an example of detecting the presence of a trace amount of a ginsenoside by double Eastern blotting and isolating and determining the structure of new ginsenosides from the American ginseng *P. quinquefolius*. The methanol extracts of American ginseng powder were analyzed by double Eastern blotting, which resulted in two minor ginsenosides, **1** and **2**. The Rf value of compound **1** was consistent with the standard ginsenoside Rc with four sugar moieties, but the reddish spot, which may be mixed with the other ginsenosides, was determined to be a protopanaxatriol-type ginsenoside. The Rf value of compound **2** was slightly lower compared with that of ginsenoside Rb1. Based on these results, compound **1** was considered a protopanaxatriol with four sugars attached to it. Meanwhile, compound **2** was determined to be a protopanaxadiol with five sugars attached. To isolate and elucidate these structures, the saponin fraction was subjected to a Diaion HP-20 column, a silica gel column, and a reversed-phase column to yield new ginsenosides designated quinquenoside Ja and quinquenoside Jb, of which the structures were determined by instrumental analyses, such as NMR, ^13^CMR, and MS spectrometry (Figure 6). In this study, 400 g of American ginseng was used for the isolation; however, if purification was performed without an Eastern blot analysis, and more samples would be needed and more labor would be required for isolation. Thus, the power of Eastern blotting was demonstrated [62].

Next, a more rigorous quality evaluation was performed by targeting two different components, GL and LQ, in licorice using two MAbs, the anti-GL MAb and anti-LQ MAb, respectively (Figure 7).

Each quantitative analysis was accurately performed by ELISA individually. The authors also developed a method to analyze both components simultaneously. The double Eastern blot analysis for GC and LQ succeeded by using double MAbs. LQ and GL were completely separated using a solvent system of 1-butanol/water/acetic acid (7/2/1, *v*/*v*/*v*). Because the color intensity of the two spots corresponded well with the ELISA values for both compounds, double Eastern blotting was used to simultaneously detect LQ and GL in licorice and to estimate the approximate content of the two compounds [63].

Next, fresh licorice root was sliced and Eastern blotting was performed using the anti-GL MAb. Areas containing GL were stained light red. Medulla and phloem parts were darker, indicating a higher GL content compared with the xylem parts. This was confirmed by measuring GL content by ELISA. However, based on double Eastern blotting, it stained purple and the distribution pattern was almost identical to that of Eastern blotting. This indicates that GL and LQ are stored at the same site. This is consistent with the results of Hayashi et al. [18,19], in which both GL and LQ selectively accumulated in the woody parts of the roots and stolon of licorice, whereas the GL content was positively correlated with the LQ content in the roots. Because only GL and LQ were detected, the presence of related compounds between GL and LQ was not determined by LC-MS or other methods, which is required for a more detailed analysis. Recently, Ochi et al., simultaneously detected GL and sennoside A in the Kampo medicine Daiokanzoto by a lateral flow immunoassay using the anti-GL MAb and anti-sennoside A MAb [64]. Although not an Eastern blot assay system, this method is useful for detecting two active ingredients in herbal medicines more easily than using two MAbs. It may be used as a general-purpose method for the quality evaluation of Kampo medicines by simultaneously detecting three or four marker ingredients.

## 4. Knockout Extract

Traditional Chinese medicine (TCM) and Kampo medicine include a combination of various herbal medicines. It is not easy to determine which herb is most effective in the prescription. To clarify which of the 13 prescribed herbal medicines of the Kampo medicine Kamiuntanto is responsible for the effects of increasing choline acetyltransferase (ChAT) activity [65,66,67], Yabe et al., prepared a formulation without one of the constituents. The activity of ChAT was not observed in the formulation without *Polygala tenuifolia* root, which indicated that it is an active herbal medicine [68]. Although the practical effects of TCM are being evaluated scientifically using clinical cases, the direct targeting factors for many formulas remain unknown. Recently, Wang et al., developed an in silico model and experimental validation method as an efficient and reliable method of assessing the target effects of formulations containing three herbal medicines [69]. Although the research to date has mainly focused on the search for medicinal components, Cai et al., studied the interactions among the components of TCM, suggesting that other components rather than the main constituents may also have important effects [70]. The authors believe that a true target analysis of TCM is possible by combining biological research and immunological technology. We attempted to elucidate the role of GL in licorice using an extract of licorice devoid of GL, which is a major component of licorice. The authors conducted this study based on the idea that by removing one major component from a crude extract and examining the activity of the remaining extract, we could determine the true role of its major component. GL, the main ingredient of licorice, was selected to prove this hypothesis. Licorice is the most important herbal medicine and is included in >70% of Kampo medicine prescriptions in Japan. The main ingredient of licorice, which contains 500 constituents, is GL, the content of which is defined as 2% or more in the Japanese Pharmacopoeia. GL has long been used as a pharmaceutical product to treat hepatitis and allergies in Japan; however, there are known side effects of GL. An overdose of GL results in pseudoaldosteronism (i.e., high blood pressure, natriuretic and fluid retention, hypocalcemia, edema, and weight gain). For this reason, it is recommended that the daily dose of licorice in Kampo medicines in complex formulas remains below 6 g [71].

The purified anti-GL MAb was treated with NaIO_4_ to yield a dialdehyde group in the sugar moiety, which was conjugated to Affi-Gel Hz (hydrazide gel), resulting in a hydrazone-type immunoaffinity gel as described above. The crude licorice root extract was loaded onto the immunoaffinity column, washed, and eluted. The washing fraction contained many components, and when compared with the Eastern blotting results, most of the components, except GL, were eluted. After complete washing, the column was eluted with MeOH to yield GL (yield: 99.6%) because of the strong binding between the antibody and its target. The washing fraction contained all of the components except GL, which was designated a GL-knockout licorice extract or a “knockout extract” [33].

Nitric oxide (NO) is involved in major central nervous system functions [72], such as increased cerebral blood flow, inhibition of atherosclerosis, prevention of thrombosis and embolism, regulation of neurotransmitter release, and synaptic plasticity [73]. Moreover, NO overexpression in the brain is thought to cause Parkinson’s Disease [74,75] and it can trigger other diseases. As mentioned above, NO is an essential molecule that has a variety of functions. In this section, we present experiments using the knockout extract with iNOS, an enzyme that biosynthesizes NO as an indicator. Macrophages produce a large amount of NO by bacterial lipopolysaccharide (LPS) stimulation, which results in cytokine production in inflammatory diseases [76]. Therefore, the authors selected murine RAW264 macrophage cells as an experimental system. The licorice extract completely inhibited NO production compared with LPS treatment. iNOS protein and mRNA expression were downregulated as confirmed by Western blotting and RT-PCR, respectively; however, GL treatment alone showed no iNOS production. When compared with the knockout and licorice extracts, the latter had a greater effect. This may be because the knockout extract did not contain GL. Therefore, we tested the knockout extract with GL (i.e., in the form of a restored licorice extract) and the inhibitory effect was enhanced, although GL alone did not inhibit NO production. This indicates that the knockout extract and GL synergistically reduce the inhibition of NO production. The authors mentioned earlier that licorice contains more than 500 constituents and it is evident that GL acts in combination with other constituents. As mentioned earlier, pseudoaldosteronism is a problem with Kampo medicines containing licorice; however, the inclusion of licorice is considered essential for its effects. One solution is to prepare a GL-knockout extract of licorice and add an appropriate amount of GL to prepare a regenerated licorice extract, which would avoid the GL side effects. Knockout extracts represent a practical method not only for research, but also for medicinal use, and should be considered in the future.

Tubulointerstitial fibrosis is a major pathologic feature of diabetic nephropathy [77]. The epithelial–mesenchymal transition (EMT) of renal proximal tubular cells plays an important role in tubulointerstitial fibrosis [63]. Therefore, to elucidate the detailed mechanisms of EMT in renal tubular cells under high-glucose (HG) conditions, the authors examined the potential of licorice to inhibit HG-induced EMT. Renal tubular epithelial cells (NRK-52E) exposed to HG exhibited EMT induction characterized by increased fibronectin and α-smooth muscle actin (α-SMA) decreasing E-cadherin. Increased levels of truncated Notch2, mastermind-like transcriptional coactivator 1 (MAML-1), nicastrin, Jagged-1, and Delta-like 1 were also evident in HG cultured cells. The overexpression of key components of Notch2 signaling in NRK-52E cells suggested that an activated Notch2 pathway, which is an important regulator of brain stem cell growth and survival [78], is essential for tubular EMT. Hsu et al., found that a licorice extract containing GL or a GL-knockout extract of licorice effectively inhibited HG-induced EMT in NRK-52E cells by suppressing the Notch2 pathway. This active constituent in licorice may include compounds other than GL. Thus, the findings suggest that Notch2-mediated renal tubular EMT is a potential therapeutic target for diabetic nephropathy [79]. Diabetic nephritis may be classified into the following stages: the early stage of nephropathy, early renal phase, manifest nephropathy stage, and renal failure and dialysis. Because GL is the causative agent of pseudohyperaldosteronism, GL-knockout extracts of licorice may be useful in diabetes research and treatment.

In this section, we summarized cell-based studies using GL-knockout extracts. Comprehensive in vivo studies, including animal models and clinical trials, are necessary to validate the pharmacological effects and safety profile of GL-knockout extracts. In the future, we hope to expand the knockout extracts to in vivo studies. Moreover, to gain a deeper understanding of the therapeutic potential and mode of action, the underlying mechanisms through which the GL-knockout extract or other constituents exert their effects need to be identified.

## 5. Conclusions

MAbs have become essential reagents in the field of biochemistry, but they have not been widely used in the natural product area. Among the studies using MAbs, the Eastern blot method has proven to be useful for the identification of known compounds and the identification and structure determination of unknown compounds. In the case of glycosides, the Rf values derived from TLC are determined and the number of bound sugars can be estimated, which is useful information for structure determination. The activity of glycosides is stronger when the number of sugars attached to the aglycon increases [59,60,61]; thus, the activity can be estimated by an Eastern blot analysis. In contrast, double Eastern blotting using two types of MAbs can isolate compounds and dye glycosides with different pharmacological activities using two different colors. This enables quality evaluation based on pharmacological activity. The extract from which one component was removed from the crude drug by using a MAb affinity column was designated a knockout extract. Thus, this method is useful in the search for active ingredients. The active fraction for diabetic nephritis was shown to be a component of a licorice extract without GL, which is the causative agent of pseudohyperaldosteronism. Thus, the appropriate dose of GL in pharmaceuticals for hepatitis and allergies in Japan should be considered.

We also discussed the use of GL-knockout extracts of licorice, because GL induces side effects [70], and caution must be taken for the overlapping and prolonged administration of Kampo medicines containing licorice. However, pure GL can be obtained through a one-step purification using the anti-GL MAb; thus, GL could be added back to extracts in appropriate amounts to create a licorice extract with no side effects. A similar herbal drug is Ephedra, in which ephedrine is the active ingredient, which is known for its antitussive, analgesic, antifebrile, and expectorant effects. However, many adverse effects of ephedrine, the main ingredient, have been observed, including excessive sweating, nausea, vomiting, and increased heart rate. Therefore, an ephedrine knockout extract of an Ephedra extract was prepared by removing ephedrine by a cation exchange column. Its anti-inflammatory and analgesic effects were confirmed [80] as well as its safety [81]. Based on these results, it may be possible to create an Ephedra extract with no side effects by completely removing ephedrine using an anti-ephedrine MAb affinity column. An appropriate amount of ephedrine could be added back to obtain the desired antitussive, analgesic, and antipyretic effects. The concept of knockout extracts is also important in terms of reducing side effects because there are many Kampo medicines, in which the main ingredient causes side effects. Therefore, it is expected that additional studies will be conducted on the practicality of MAbs as a further application.

MAbs enable the rapid detection of active compounds on the membrane and reveal previously unknown roles of active compounds. In addition, they facilitate the study of the potential function of a target compound in complex phytochemical mixtures. Furthermore, the cellular localization and molecular targets of active compounds can be analyzed to clarify the underlying mechanisms using MAb-based molecular techniques. The applications of MAbs against natural compounds will further expand the functional analysis of natural products and their therapeutic applications. We hope that this review will stimulate research and development to expand the diversity of MAbs that target various active compounds, particularly those relevant to therapeutic applications.

## Figures and Tables

**Figure 1 antibodies-13-00060-f001:**
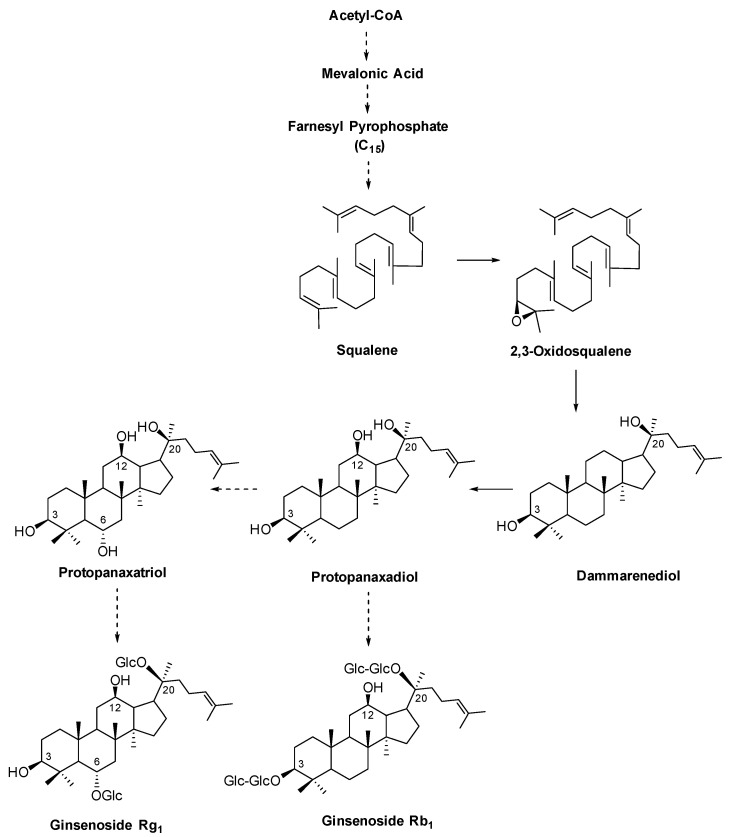
Biosynthetic pathway of ginsenosides [5].

**Figure 2 antibodies-13-00060-f002:**
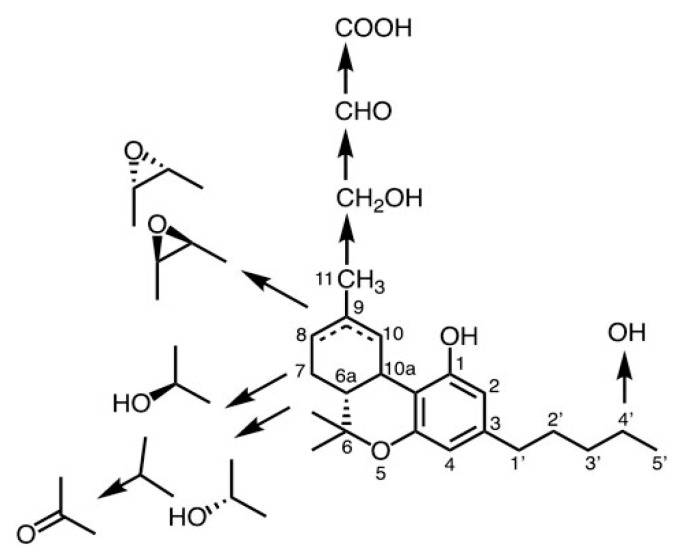
Metabolic map for THC [27].

**Figure 3 antibodies-13-00060-f003:**
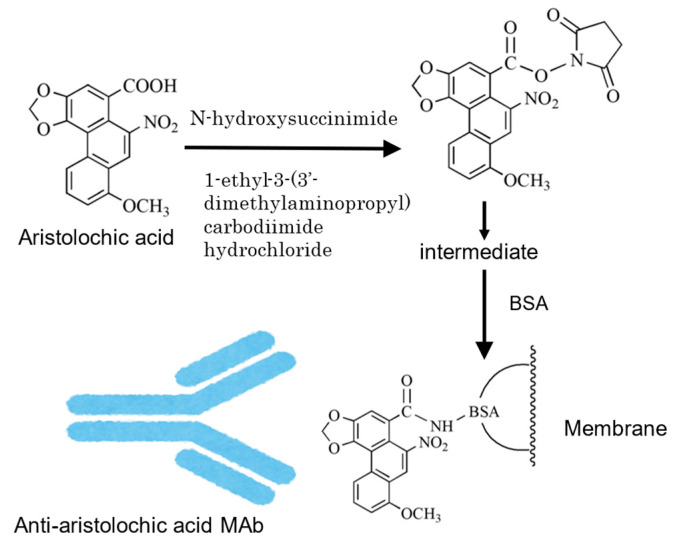
Scheme of Eastern blotting for compounds without sugar [41].

**Figure 4 antibodies-13-00060-f004:**
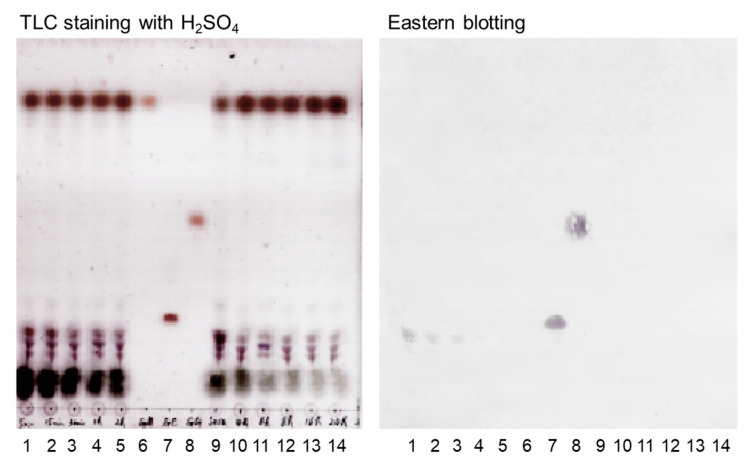
Experimental administration of glycyrrhizin in mice. **1**: 5 min, **2**: 15 min, **3**: 30 min, **4**: 1 h, **5**: 2 h, **6**: GA, **7**: GL, **8**: GA monoglucronide, **9**: serum standard, **10**: 4 h, **11**: 6 h, **12**: 8 h, **13**: 16 h, and **14**: 24 h.

**Figure 5 antibodies-13-00060-f005:**
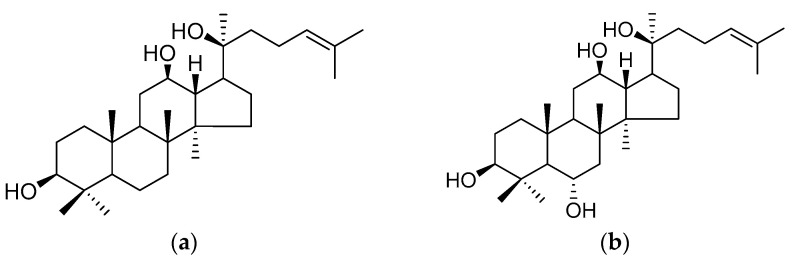
Structures of protopanaxadiol (**a**) and protopanaxatriol (**b**).

**Figure 6 antibodies-13-00060-f006:**
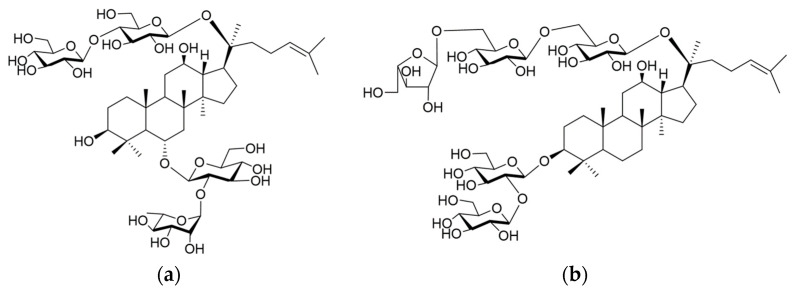
Structures of two newly isolated ginsenosides, quinquenoside Ja (**a**) and quinquenoside Jb (**b**), from American ginseng.

**Figure 7 antibodies-13-00060-f007:**
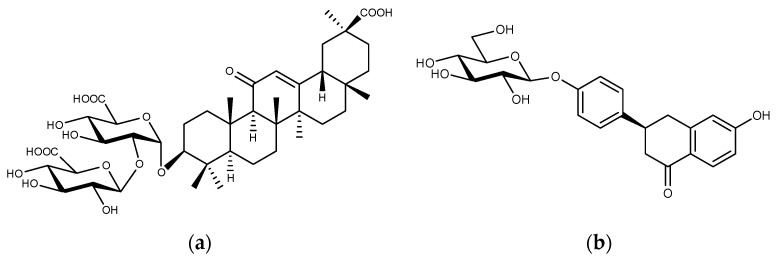
Structures of GL (**a**) and LQ (**b**).

**Table 1 antibodies-13-00060-t001:** Analysis of ginsenoside Rb1 in Araliaceous plants using an anti-ginsenoside Rb1 monoclonal antibody. (− : undetectable, + : detectable, ++ : strongly detectable).

	ELISA	Eastern Blotting
1. *Acanthopanax japonicus*	−	−
2. *A. spinosus*	−	−
3. *A. sieboldianus*	+	−
4. *A. divaricatus*	−	−
5. *A hypoleucus*	−	−
6. *A. sciadophylloides*	−	−
7. *Eleutherococcus senticosus*	−	−
8. *Aralia elata*	+	−
9. *A. cordata*	+	−
10. *Dendropanax trifidus*	−	−
11. *Faysia japonica*	+	+
12. *Kalopanax pictus*	++	++

## Data Availability

No new data were created or analyzed in this study. Data sharing is not applicable to this article.

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
