# Peer review of "Application of Monoclonal Antibodies against Naturally Occurring Bioactive Ingredients"

_2073-4468, 2024, doi:10.3390/antib13030060_

Round 1
Reviewer 1 Report
Comments and Suggestions for Authors
Comments to the Author:
The study explores the use of monoclonal antibodies (MAbs) in identifying naturally occurring active compounds, focusing on glycosides and their pharmacological effects. It introduces Eastern blotting as a method to visually represent specific components of lower molecular weight compounds, especially glycosides, using MAbs. The study also discusses the concept of "knockout extract" derived from MAbs-associated affinity columns and its potential synergistic activity with active compounds like glycyrrhizin (GL) in inhibiting high glucose-induced epithelial-mesenchymal transition, suggesting implications for diabetic nephritis.
The manuscript is not suitable for publication in the present format. May be considered for publication after extensive re-writing in response to the below given comments and queries.
The study has several limitations and drawbacks that should be considered when interpreting its findings:
The major points that needs to be addressed are mentioned below:
- There is a lot of grammatical and punctuation mistakes. Please check your manuscript from a native English speaker or give it to a professional English editing services.
2. Please provide some more latest references in the manuscript.
3. The study mentions less than 50 types of MAbs to active compounds, indicating a potential limitation in the diversity of MAbs available for various compounds. Encourage further research and development to expand the diversity of MAbs targeting different active compounds, especially those relevant to therapeutic applications.
4. The findings regarding the synergistic activity of GL-knockout extract in inhibiting epithelial-mesenchymal transition are based on cell culture models and may not fully represent in vivo complexities or clinical outcomes.
5. While the study suggests potential implications for diabetic nephritis, further in vivo studies and clinical trials would be needed to validate the efficacy and safety of GL-knockout extract in humans.
6. Conduct comprehensive in vivo studies, including animal models and potentially clinical trials, to validate the pharmacological effects and safety profile of GL-knockout extract or similar formulations in treating diabetic nephritis or related conditions.
7. Investigate the underlying mechanisms by which GL-knockout extract or other constituents exert their effects, such as through the Notch2 pathway, to provide a deeper understanding of their therapeutic potential and mode of action.
Comments on the Quality of English Language
- There is a lot of grammatical and punctuation mistakes. Please check your manuscript from a native English speaker or give it to a professional English editing services.
Author Response
We are very much thankful to the editor and reviewers for their deep and thorough review. We have revised our manuscript in the light of their useful suggestions and comments. We hope our revision has improved the paper to a level of their satisfaction. Listed below are our answers to their comments.
Reviewer #1:
General comment: The study explores the use of monoclonal antibodies (MAbs) in identifying naturally occurring active compounds, focusing on glycosides and their pharmacological effects. It introduces Eastern blotting as a method to visually represent specific components of lower molecular weight compounds, especially glycosides, using MAbs. The study also discusses the concept of "knockout extract" derived from MAbs-associated affinity columns and its potential synergistic activity with active compounds like glycyrrhizin (GL) in inhibiting high glucose-induced epithelial-mesenchymal transition, suggesting implications for diabetic nephritis.
The manuscript is not suitable for publication in the present format. May be considered for publication after extensive re-writing in response to the below given comments and queries.
Response: Thank you for the comments to improve our manuscript. In accordance with the reviewer's comments, we have added several information and re-written our manuscript as shown below.
Major comments:
Comment: 1. There is a lot of grammatical and punctuation mistakes. Please check your manuscript from a native English speaker or give it to a professional English editing services.
Response: Thank you for your thoughtful review of our manuscript. As you suggested, we have had the manuscript rewritten by an experienced scientific editor, who has improved the grammar and stylistic expression of the paper. We have added the following text in “Acknowledments” (page 16, lines 574-575).
(Page 16, lines 574-575)
The authors would like to thank Enago (www.enago.jp) for the English language review.
Comment: 2. Please provide some more latest references in the manuscript.
Response: In accordance with the reviewer’s comment, we added latest references to the manuscript.
(References)
- Liu, L.; Xu, FR.; Wang, YZ. Traditional uses, chemical diversity and biological activities of Panax L. (Araliaceae): A review. J. Ethnopharmacol. 2020, 263, 112792.
- Ito, H.; Ito, M. Recent trends in ginseng research. J. Nat. Med. 2024, 78, 455-466.
- Yoshino, T.; Shimada, S.; Homma, M.; Makino, T.; Mimura, M.; Watanabe, K. Clinical risk factors of licorice-induced pseudoaldosteronism based on glycyrrhizin-metabolite concentrations: A narrative review. Front. Nutr. 2021, 8, 719197.
- Blebea, NM.; Pricopie, AI.; Vlad, RA.; Hancu, G. Phytocannabinoids: Exploring pharmacological profiles and their impact on therapeutical use. Int. J. Mol. Sci. 2024, 25, 4204.
- Iova, OM.; Marin, GE.; Lazar, I.; Stanescu, I.; Dogaru, G.; Nicula, CA.; Bulboacă, AE. Nitric oxide/nitric Oxide Synthase dystem in the pathogenesis of neurodegenerative disorders-An overview. Antioxidants (Basel) 2023, 12, 753.
- Xu, C.; Ha, X.; Yang, S.; Tian, X.; Jiang, H. Advances in understanding and treating diabetic kidney disease: focus on tubulointerstitial inflammation mechanisms. Front. Endocrinol. (Lausanne), 2023, 14, 1232790.
- Hadpech, S.; Thongboonkerd, V. Epithelial-mesenchymal plasticity in kidney fibrosis. Genesis 2024, 62, e23529.
Comment: 3. The study mentions less than 50 types of MAbs to active compounds, indicating a potential limitation in the diversity of MAbs available for various compounds. Encourage further research and development to expand the diversity of MAbs targeting different active compounds, especially those relevant to therapeutic applications.
Response: We thank the reviewer for this encouraging comments. We have added the following text to this section (page 16, lines 563-565).
(Page 16, lines 563-565)
We hope that this review will stimulate research and development to expand the diversity of MAbs that target various active compounds, particularly those relevant to therapeutic applications.
Comment: 4. The findings regarding the synergistic activity of GL-knockout extract in inhibiting epithelial-mesenchymal transition are based on cell culture models and may not fully represent in vivo complexities or clinical outcomes.
5. While the study suggests potential implications for diabetic nephritis, further in vivo studies and clinical trials would be needed to validate the efficacy and safety of GL-knockout extract in humans.
6. Conduct comprehensive in vivo studies, including animal models and potentially clinical trials, to validate the pharmacological effects and safety profile of GL-knockout extract or similar formulations in treating diabetic nephritis or related conditions.
Response: We appreciate the reviewer's concerns on these points. Our previous studies using GL-knockout extract were performed on cell-based studies. As reviewer noted, the comprehensive in vivo studies, including animal models and potentially clinical trials, are necessary to validate the pharmacological effects and safety profile of GL-knockout extract. In the future, we will expand the research using knockout extracts to in vivo studies. Thus, we have added the following text to the end of section “4. Knockout extract” (page 15, lines 515-518). We believe that this new information adequately addresses the Reviewer's comment.
(Page 15, lines 515-518)
In this section, we summarized cell-based studies using GL-knockout extracts. Comprehensive in vivo studies, including animal models and clinical trials, are necessary to validate the pharmacological effects and safety profile of GL-knockout extracts. In the future, we hope to expand the knockout extracts to in vivo studies.
Comment: 7. Investigate the underlying mechanisms by which GL-knockout extract or other constituents exert their effects, such as through the Notch2 pathway, to provide a deeper understanding of their therapeutic potential and mode of action.
Response: Thank you for useful comment to improve our manuscript. In accordance with the reviewer’s comment, we have added the following text to the end of section “4. Knockout extract” (page 15, lines 518-521).
(Page 15, lines 518-521)
Moreover, to gain a deeper understanding of the therapeutic potential and mode of action, the underlying mechanisms through which GL-knockout extract or other constituents ex-ert their effects need to be identified.
Reviewer 2 Report
Comments and Suggestions for Authors
In this review, Fujii et al, explained the use of Mab in different application other than therapeutic use. The authors have explained how MAb are used for the isolation of the active ingredients using the column from the natural extract called knockout extract to reduce the side effects which is the very useful technique for reducing the side effects.
The authors are also focused on the eastern blotting and double eastern blotting method where Mab for the active ingredients are used to determine the structure and activity of the of the active ingredient are determined.
This review is useful for the studies to develop natural medicines for the determination of the effect of active ingredient and to minimize the side effects of it.
No Major Comments
Author Response
We are very much thankful to the editor and reviewers for their deep and thorough review. We have revised our manuscript in the light of their useful suggestions and comments. We hope our revision has improved the paper to a level of their satisfaction. Listed below are our answers to their comments.
Reviewer #2:
General comment: The authors have explained how MAb are used for the isolation of the active ingredients using the column from the natural extract called knockout extract to reduce the side effects which is the very useful technique for reducing the side effects.
The authors are also focused on the eastern blotting and double eastern blotting method where Mab for the active ingredients are used to determine the structure and activity of the of the active ingredient are determined.
This review is useful for the studies to develop natural medicines for the determination of the effect of active ingredient and to minimize the side effects of it.
Response: We are grateful to the reviewer for the positive and encouraging comments.
Reviewer 3 Report
Comments and Suggestions for Authors
This paper explores the use of monoclonal antibodies (MAbs) to target naturally occurring active compounds. It highlights Eastern blotting for detection and MAb-based affinity columns for isolation, while showing licorice research as an example.
My comments and concerns:
1. The introduction provides a comprehensive overview of MAbs' applications in various fields, including natural product chemistry and Kampo medicine.
2. The manuscript describes various methods, such as eastern blotting and the creation of knockout extracts, in detail.
3. The results section details the application of MAbs in identifying and quantifying bioactive compounds in natural products. The discussion on the synergistic effects of glycyrrhizin (GL) and knockout extracts is intriguing, but would benefit from a deeper analysis of how these findings advance antibody research specifically at the abstract and conclusion.
4. The manuscript contains inconsistencies in writing style, alternating between first-person ("we") and third-person ("the authors") perspectives. For coherence and professionalism, it is recommended to choose one style and apply it consistently throughout the manuscript.
5. The figures and tables are well-organized and provide clear representations of the experimental results.
6. The references are appropriate and encompass a wide range of relevant literature.
7. The conclusion effectively summarizes the key findings but should emphasize their implications for antibody research and potential future applications in this field.
Overall, manuscript offers valuable insights into the use of MAbs in natural product research, but its relevance to the scope of MDPI Antibodies journal may be limited. The authors might consider submitting to a journal more focused on natural products or pharmacognosy or check with the editor and revise it accordingly as needed.
Comments on the Quality of English LanguageConsistency in writing style should also be addressed.
Author Response
We are very much thankful to the editor and reviewers for their deep and thorough review. We have revised our manuscript in the light of their useful suggestions and comments. We hope our revision has improved the paper to a level of their satisfaction. Listed below are our answers to their comments.
Reviewer #3:
Comment: 1. The introduction provides a comprehensive overview of MAbs' applications in various fields, including natural product chemistry and Kampo medicine.
2. The manuscript describes various methods, such as eastern blotting and the creation of knockout extracts, in detail.
Response: We are grateful to the reviewer for the positive comments.
Comment: 3. The results section details the application of MAbs in identifying and quantifying bioactive compounds in natural products. The discussion on the synergistic effects of glycyrrhizin (GL) and knockout extracts is intriguing, but would benefit from a deeper analysis of how these findings advance antibody research specifically at the abstract and conclusion.
Response: We thank the reviewer for this comment. In accordance with the reviewer’s comment, we have added the following text as shown below.
(Page 15, lines 515-521)
In this section, we summarized cell-based studies using GL-knockout extracts. Com-prehensive in vivo studies, including animal models and clinical trials, are necessary to validate the pharmacological effects and safety profile of GL-knockout extracts. In the fu-ture, we hope to expand the knockout extracts to in vivo studies. Moreover, to gain a deep-er understanding of the therapeutic potential and mode of action, the underlying mecha-nisms through which GL-knockout extract or other constituents exert their effects need to be identified.
(Page 16, lines 559-563)
Furthermore, the cellular localization and molecular targets of active compounds can be analyzed to clarify the underlying mechanisms using MAb-based molecular techniques. The applications of MAbs against natural compounds will further expand the functional analysis of natural products and their therapeutic applications.
Comment: 4. The manuscript contains inconsistencies in writing style, alternating between first-person ("we") and third-person ("the authors") perspectives. For coherence and professionalism, it is recommended to choose one style and apply it consistently throughout the manuscript.
Response: In accordance with the reviewer’s comment, we have revised the manuscript by using the consistent terminology.
Comment: 5. The figures and tables are well-organized and provide clear representations of the experimental results.
6. The references are appropriate and encompass a wide range of relevant literature.
Response: We are grateful to the reviewer for the positive comments. .In order to further improve our manuscript, we added latest references to the manuscript.
(References)
- Liu, L.; Xu, FR.; Wang, YZ. Traditional uses, chemical diversity and biological activities of Panax L. (Araliaceae): A review. J. Ethnopharmacol. 2020, 263, 112792.
- Ito, H.; Ito, M. Recent trends in ginseng research. J. Nat. Med. 2024, 78, 455-466.
- Yoshino, T.; Shimada, S.; Homma, M.; Makino, T.; Mimura, M.; Watanabe, K. Clinical risk factors of licorice-induced pseudoaldosteronism based on glycyrrhizin-metabolite concentrations: A narrative review. Front. Nutr. 2021, 8, 719197.
- Blebea, NM.; Pricopie, AI.; Vlad, RA.; Hancu, G. Phytocannabinoids: Exploring pharmacological profiles and their impact on therapeutical use. Int. J. Mol. Sci. 2024, 25, 4204.
- Iova, OM.; Marin, GE.; Lazar, I.; Stanescu, I.; Dogaru, G.; Nicula, CA.; Bulboacă, AE. Nitric oxide/nitric Oxide Synthase dystem in the pathogenesis of neurodegenerative disorders-An overview. Antioxidants (Basel) 2023, 12, 753.
- Xu, C.; Ha, X.; Yang, S.; Tian, X.; Jiang, H. Advances in understanding and treating diabetic kidney disease: focus on tubulointerstitial inflammation mechanisms. Front. Endocrinol. (Lausanne), 2023, 14, 1232790.
- Hadpech, S.; Thongboonkerd, V. Epithelial-mesenchymal plasticity in kidney fibrosis. Genesis 2024, 62, e23529.
Comment: 7. The conclusion effectively summarizes the key findings but should emphasize their implications for antibody research and potential future applications in this field.
Response: In accordance with the reviewer’s comment, we have added the following text as shown below.
(Page 16, lines 557-565)
MAbs enable the rapid detection of active compounds on the membrane and reveal previously unknown roles of active compounds. In addition, they facilitate the study of the potential function of a target compound in complex phytochemical mixtures. Furthermore, the cellular localization and molecular targets of active compounds can be analyzed to clarify the underlying mechanisms using MAb-based molecular techniques. The applications of MAbs against natural compounds will further expand the functional analysis of natural products and their therapeutic applications. We hope that this review will stimulate research and development to expand the diversity of MAbs that target various active compounds, particularly those relevant to therapeutic applications.
Comment: Overall, manuscript offers valuable insights into the use of MAbs in natural product research, but its relevance to the scope of MDPI Antibodies journal may be limited. The authors might consider submitting to a journal more focused on natural products or pharmacognosy or check with the editor and revise it accordingly as needed.
Response: We thank the reviewer for this comment. We leave the final decision to the editor regarding whether to accept in the “antibodies”.
Round 2
Reviewer 1 Report
Comments and Suggestions for Authors
Manuscript can be accepted in the present format.
Reviewer 3 Report
Comments and Suggestions for Authors
Authors have satisfactorily addressed my concerns and points.